# What is meant by validity in maternal and newborn health measurement? A conceptual framework for understanding indicator validation

**Lenka Benova**[1,2]*, **Ann-Beth Moller**[3], **Kathleen Hill**[4], **Lara M. E. Vaz**[5], **Alison Morgan**[6], **Claudia Hanson**[7,8], **Katherine Semrau**[9], **Shams Al Arifeen**[10], **Allisyn C. Moran**[11]

1 Faculty of Epidemiology and Population Health, London School of Hygiene and Tropical Medicine, London, United Kingdom, 2 Department of Public Health, Institute of Tropical Medicine, Antwerp, Belgium, 3 Department of Sexual and Reproductive Health and Research, UNDP/UNFPA/UNICEF/WHO/World Bank Special Programme of Research, Development and Research Training in Human Reproduction (HRP), Department of Reproductive Health and Research, World Health Organization, Geneva, Switzerland, 4 Maternal Child Survival Program, Jhpiego, Washington, DC, United States of America, 5 Population Reference Bureau, Washington, DC, United States of America, 6 Nossal Institute for Global Health, University of Melbourne, Melbourne, Australia, 7 Global Public Health, Karolinska Institutet, Stockholm, Sweden, 8 Department of Disease Control, London School of Hygiene and Tropical Medicine, London, United Kingdom, 9 Division of Global Health Equity Brigham & Women's Hospital, Department of Medicine, Ariadne Labs, Harvard Medical School, Boston, Massachusetts, United States of America, 10 Maternal and Child Health Division, International Centre for Diarrhoeal Disease Research, Bangladesh (icddr,b), Dhaka, Bangladesh, 11 Department of Maternal, Newborn, Child and Adolescent Health, World Health Organization, Geneva, Switzerland

* LBenova@itg.be

**Data Availability Statement:** The data in the form of interview notes is available open access under the DOI: https://doi.org/10.17037/DATA.00001403.

## Abstract

### Background

Rigorous monitoring supports progress in achieving maternal and newborn mortality and morbidity reductions. Recent work to strengthen measurement for maternal and newborn health highlights the existence of a large number of indicators being used for this purpose. The definitions and data sources used to produce indicator estimates vary and challenges exist with completeness, accuracy, transparency, and timeliness of data. The objective of this study is to create a conceptual overview of how indicator validity is defined and understood by those who develop and use maternal and newborn health indicators.

### Methods

A conceptual framework of validity was developed using mixed methods. We were guided by principles for conceptual frameworks and by a review of the literature and key maternal and newborn health indicator guidance documents. We also conducted qualitative semi-structured interviews with 32 key informants chosen through purposive sampling.

### Results

We categorised indicator validity into three main types: criterion, convergent, and construct. Criterion or diagnostic validity, comparing a measure with a gold standard, has

**Funding:** This work received support from the Bill & Melinda Gates Foundation.

**Competing interests:** The authors have declared that no competing interests exist.

predominantly been used to assess indicators of care coverage and content. Studies assessing convergent validity quantify the extent to which two or more indicator measurement approaches, none of which is a gold-standard, relate. Key informants considered construct validity, or the accuracy of the operationalisation of a concept or phenomenon, a critical part of the overall assessment of indicator validity.

## Conclusion

Given concerns about the large number of maternal and newborn health indicators currently in use, a more consistent understanding of validity can help guide prioritization of key indicators and inform development of new indicators. All three types of validity are relevant for evaluating the performance of maternal and newborn health indicators. We highlight the need to establish a common language and understanding of indicator validity among the various global and local stakeholders working within maternal and newborn health.

## Introduction

Globally, the latest estimates indicate that 295,000 maternal deaths occurred in 2017, 2.5 million newborns died in 2018, and 2.6 million stillbirths occurred in 2015. [1–3] Tackling this burden has been prioritised in national, regional and global actions, with ambitious targets set for maternal and newborn survival and well-being. [4, 5] A range of indicators are currently used at global, regional, national and sub-national levels to monitor the progress toward these goals, including the state of maternal and newborn health and well-being, as well as the health systems and care processes thought to influence health outcomes. Various maternal and newborn health initiatives have produced core indicator lists and a recent effort to map these various indicators found a rapidly expanding number of indicators numbering over 140. [6] Data sources, methods and definitions for estimating these indicators vary and change over time, and additional challenges exist with completeness, accuracy, transparency, and timeliness of available data.

For indicators to track progress, they must be measurable and clearly defined, accurate, reliable, valid, useful, relevant, accessible, specific, and time-bound. [7] The performance of indicators used for global monitoring along these dimensions is of crucial concern. Within the field of maternal and newborn health, work on measuring and improving validity of currently used indicators and indicators under development is a key part of this agenda. [8–11] Assessing the scientific robustness of indicators in the field of maternal and newborn health goes back several decades, along with development of measurement methods. More recently, several high-profile global efforts to identify and prioritise the most relevant maternal and newborn health indicators for consistent and up-to-date tracking of progress have resulted in additional research on indicator validity. [12, 13]

Given the amount of ongoing work to strengthen measurement for maternal and newborn health, increased coordination and harmonization of efforts are essential. [14] Maternal and newborn health are inextricably linked and it is important that measurement efforts address both maternal and newborn health, capture stillbirths, and other perinatal outcomes. In 2015, the World Health Organization (WHO) launched the Mother and Newborn Information for Tracking Outcomes and Results Technical Advisory Group (MoNITOR), which functions as a Technical Advisory body to the WHO on matters of measurement, metrics, and monitoring of maternal and newborn health for the Departments of Maternal, Newborn, Child and

Adolescent Health and Reproductive Health and Research. [15, 16] The purpose of MoNITOR is to provide clear, independent, harmonized, and strategic advice for global and country stakeholders engaged in maternal and newborn health measurement and accountability. This paper is a result of research commissioned and chaired by the MoNITOR Secretariat to provide global guidance.

## Objective

The objective of this paper is to present a range of perspectives on how validity of maternal and newborn indicators is defined, understood, and measured by those who develop and use these indicators. We define validity as the level of scientific robustness of an indicator with respect to how well it captures a phenomenon or concept of interest. [17] We focus on the overall meaning of indicator validity, that is, the extent to which an indicator correctly measures an underlying maternal and newborn health phenomenon. [7, 18]

We do not aim address the topic of maternal and newborn indicator validity exhaustively; rather, we concentrate on identifying common conceptual and methodological themes and provide examples of different types of validation research approaches. We focus primarily on indicators related to the Sustainable Development Goals (SDGs), [5] the Global Strategy for Women's, Children's, and Adolescents' Health, [19] Every Newborn Action Plan, [20] and Ending Preventable Maternal Mortality [21] and consider maternal and newborn health indicator validation work in countries of all income levels. However, examples are taken mainly from validation research in low- and middle-income country (LMIC) settings, as that is where the double burden of maternal and newborn morbidity and mortality as well as uncertainties regarding data quality concentrate. This framework is a part of a larger body of work led by MoNITOR to develop implementation support tools on 1. measuring validity of maternal and newborn health indicators; 2. prioritising indicators best suited for monitoring progress in various settings; 3. improving indicator usefulness and uptake by the various global and national stakeholders; and 4. identifying gaps that require additional research. These implementation support tools will also include an online tool to facilitate indicator use and interpretation.

## Materials and methods

We were guided by principles for iterative development conceptual frameworks outlined by Jabareen. [22] They propose that a conceptual framework is based on multidisciplinary bodies of knowledge, and consist of "interlinked concepts that together provide a comprehensive understanding".

We iteratively moved between data collection and analysis, starting with mapping of data sources, analysis and categorisation of selected data, identification and naming of concepts (in light of the multidisciplinary literature on validity and reliability), and integration of concepts. Between December 2017 and April 2019, we used three data gathering approaches to develop this framework. We conducted interviews with key informants, a review of the published literature [23, 24], and a review of key indicator guidance documents, which were used to construct a framework of typologies of validation studies and provide examples of various types of indicator validation work. The validation phase of constructing this conceptual framework consisted of presentations and discussion of drafts of this framework during the May 2018, November 2018, and April 2019 meetings of MoNITOR and during several meetings with MoNITOR's co-chairs, whose feedback was incorporated in this document.

The full methods and results of the key informant interviews are reported in a separate paper. [25] We used purposive sampling to identify key informants until thematic saturation was achieved. First, AM, A-BM and LB drew up a list of potential key informants through

discussion and with input from the MoNITOR co-chairs. The list was further expanded using snowball methods to encompass qualitative and quantitative measurement experts on the various types of maternal and newborn indicators (health system and input, care access and availability, quality of care and safety, coverage and outcomes, and health impact). The final sample of 32 key informants interviewed included 22 measurement experts based in academic institutions, four from funders operating in the space of maternal and newborn health, two from United Nations agencies, two from implementing agencies, and two from data collection organisations.

We used a semi-structured interview guide, pre-tested on the first five informants, covering five themes: the meaning of indicator validity, methodological approaches to assessing validity, acceptable levels of indicator validity, gaps in validation research, and recommendations for addressing these gaps. Interviews (six in person and 25 by phone/Skype) were conducted by LB in English between December 2017 and November 2018 and ranged between 45 and 90 minutes. Detailed notes were taken in shorthand during the interviews, and were transcribed and expanded immediately following the interview. Several key informants sent additional written materials (reports, unpublished manuscripts) and publications following their interview. These were included in the literature review if relevant to the study. We used the thematic content approach to analyze the interview notes and identify themes through a coding framework using a mix of deductive and inductive codes. No ethics approval was sought. All key informants were asked to review their interview notes and agreed to have their anonymized interview notes included in an open access data file. [26]

We reviewed the literature with a focus on identifying a range of study designs relevant to indicator validation within the field of maternal and newborn health. We used a combination of text and MeSH terms related to the concepts of 1. validity (validation, validity, reliability, sensitivity, specificity, verification, concordance, area under the curve, receiver operating curve), 2. maternal and newborn health (maternal, pregnancy, antenatal, childbirth, peripartum, intrapartum, labour, newborn, neonatal, postpartum, postnatal, perinatal, obstetric, stillbirth), and 3. indicators (indicator, estimate) and searched Medline, Embase, and Global Health databases on March 16, 2018 for English language articles published since 1990. Further, we used key informant recommendations of publications and reports to complement the search results. We screened the titles and abstracts of identified references (10,974 from Medline, 14,696 from Embase, 2,476 from Global Health, and 53 received from key informants). We included 119 references in full-text and used these in the development of the conceptual framework or as examples of validation studies. Last, we reviewed 12 key indicator guidance documents relevant to maternal and newborn health. [6, 8, 27–36]

## Definitions

An indicator is a quantifiable characteristic of a defined population which has a standard definition. [35, 36] We limit our consideration to indicators related to the health status and the health care of women and newborns during pregnancy, childbirth and the postnatal period. We aimed to synthesise the various perspectives on understanding and assessing validity of maternal and newborn health indicators obtained from the literature and key informant interviews and to characterise these approaches using a common language to aid efforts to achieve standard measurement language. To help characterize the various approaches used to assess validity of maternal and newborn health indicators, we classified the key types of maternal and newborn health indicators currently in use. For the purpose of this paper, we categorize indicators (Fig 1) using a framework adapted from Moller and colleagues [6] into the following key domains of maternal and newborn health indicators:

| Indicator domain | Main data source(s) | Examples of indicators | Detail of an example indicator* (N-numerator, D-denominator) |
|---|---|---|---|
| 1 Health system | Country-level informants (MoH, MoF, MoE) Policy documents Household expenditure surveys | • Newborn lifesaving commodities in essential medicine list • Birth registration • Costed national implementation plan for maternal, newborn and child health | **Newborn lifesaving commodities in essential medicine list** Number of the four listed commodities that are included in the essential medicine list (injectable antibiotics, antenatal corticosteroids, chlorhexidine and resuscitation equipment). |
| 2 Access to and availability of care | Facility surveys/ censuses Facility records (stock cards) HMIS | • Availability of bag and mask for newborn resuscitation • Availability of functional EmONC facilities | **Availability of bag and mask for newborn resuscitation** N: Number of facilities with a functional neonatal bag and two masks (sizes 0 and 1) in the labour and delivery service area D: Total number of facilities with inpatient maternity services that were assessed |
| 3 Care coverage | Population-level surveys Facility records/ observations HMIS | • Kangaroo mother care • Newborn resuscitation • Antenatal care (at least 4 visits) • Institutional (facility-based) delivery | **Kangaroo mother care** N: Number of newborns initiated on facility-based kangaroo mother care D: Target population for coverage: total number of newborns with birthweight <2000g or <2500g |
| 4 Care content and quality | Facility surveys/ censuses Facility records/ observations User surveys | • Antenatal care: blood pressure measured • Person-centeredness of care • Labor and Delivery Quality of Care Short Observational Index • Respectful maternity care | **Antenatal care: blood pressure measured** N: Number of women who attended antenatal care who had blood pressure measured D: Number of women who attended antenatal care (denominator definition might vary depending on data source) |
| 5 Impact | Population-level surveys Civil registration vital statistics Facility records HMIS | • Maternal mortality ratio (per 100,000 live births) • Stillbirth rate (per 1,000 total births) • Neonatal mortality rate (per 1,000 live births) • Percentage of women aged 20–24 years who gave birth before age 18 | **Maternal mortality ratio (per 100,000 live births)** N: Number of maternal deaths during a given time period D: Live births during the same time period x 100,000 |

* Indicator definitions may vary by data source and indicator list used.

*Notes:* EmONC: Emergency obstetric and newborn care; HMIS: health management information system (i.e. aggregate, routinely collected health service data); MoH: Ministry of Health; MoF; Ministry of Finance; MoE: Ministry of Education.

**Fig 1. Key domains of maternal and newborn health indicators.**

1. Health system–includes human and financial resources, policies, guidelines, mechanisms, and information flows.

2. Access to and availability of care—refers to accessibility of care to users, availability of health facilities, services and essential supplies and equipment.

3. Care coverage—indicators of the extent to which care is used (e.g. antenatal care and newborn care).

4. Care content and quality—includes care content (elements of care delivered as part of care processes) and person-centeredness of care.

5. Impact–refers to the long-term effects on health status, including morbidity and mortality.

An appraisal of an indicator's validity requires theoretical clarity about the concept that the indicator is intended to measure, and should be done in conjunction with an assessment of its reliability, and potentially also the feasibility of its production. Reliability, a key concept closely related to validity, captures the extent to which results are repeatable; in other words, how well the method is able to achieve similar measurement over repeated efforts. [36, 37] Studies in the field of maternal and newborn indicators assessing reliability also use the terms consistency, agreement, and concordance; studies assessing reliability of measures over time also use the terms decay/deterioration (of recall), and repeatability.

The four scenarios of the combination of high/low criterion validity and reliability of a measurement are visualised in Fig 2. The center of the bullseye represents the truth or the gold standard against which criterion validity is assessed while the dots represent data points. [38] As can be seen in the scenarios, consistent (reliable) indicator measurement may or may not be accurately capturing the "truth" or gold standard, while consistently valid measurement (hitting the bullseye) may still result in broad variations in estimates (limited reliability). The possibility of an indicator measurement having relatively low reliability yet still being valid differs from the perspective of other social science disciplines; it is a result of a situation where measurement is not precise on an individual level, but without systematic bias, and this produces estimates close to the truth on a population level (captured, for example, by inflation factor). [39–42]

## Results

Three main types of validity of maternal and newborn health indicators were identified from the existing literature and key informant interviews (S1 Table). These types broadly map onto the social science definitions of criterion, convergent, and construct validity. Fig 3 shows an example of the three types of validity in relation to one construct and two potential indicators measuring this construct. We describe each type of indicator validity in detail, giving examples of indicators and published studies, with a focus on approaches and measurement methods used to assess validity.

a. <u>Criterion validity</u>: Assessment of criterion validity, also referred to as diagnostic validity, examines whether the operationalization or measurement of a construct behaves as expected. A common way to examine criterion validity is to compare a measurement with a "gold-standard" or reference standard.

b. <u>Convergent validity</u>: Assessments of convergent validity examine the extent to which one measurement is similar to (converges with) other measurements to which it should be related, based on a common underlying construct (i.e. assessment of different methods of

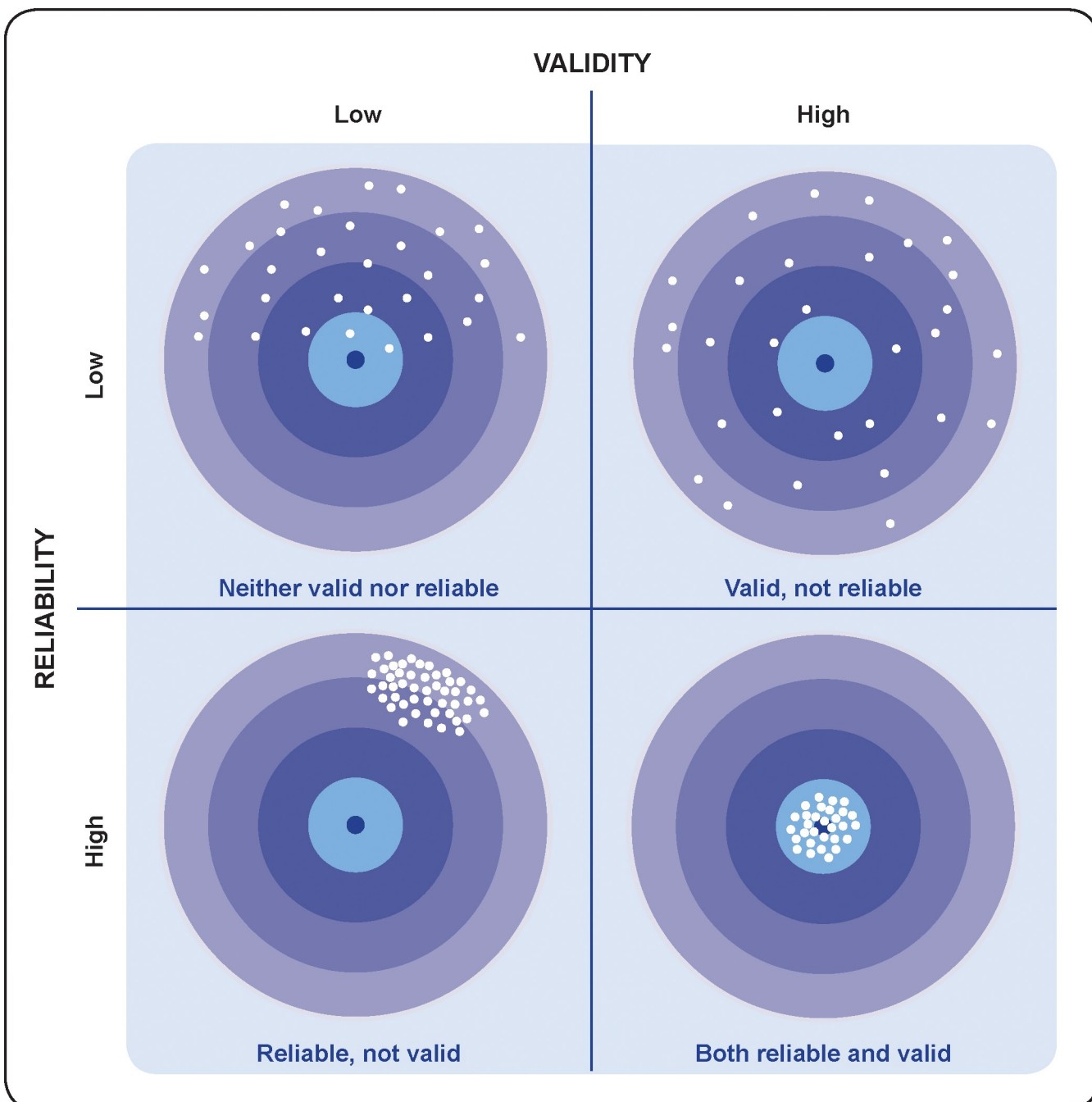

**Fig 2. Visual representation of criterion validity and reliability of measurement.**

capturing the same construct). The main difference between criterion and convergent validity is that for the second, no gold standard measurement is available, which is why new or indirect measures are sometimes referred to as surrogate or proxy indicators. Assessments of convergent validity in maternal and newborn health have compared two or more indicators, or two or more measurement methods to estimate one indicator (Fig 3).

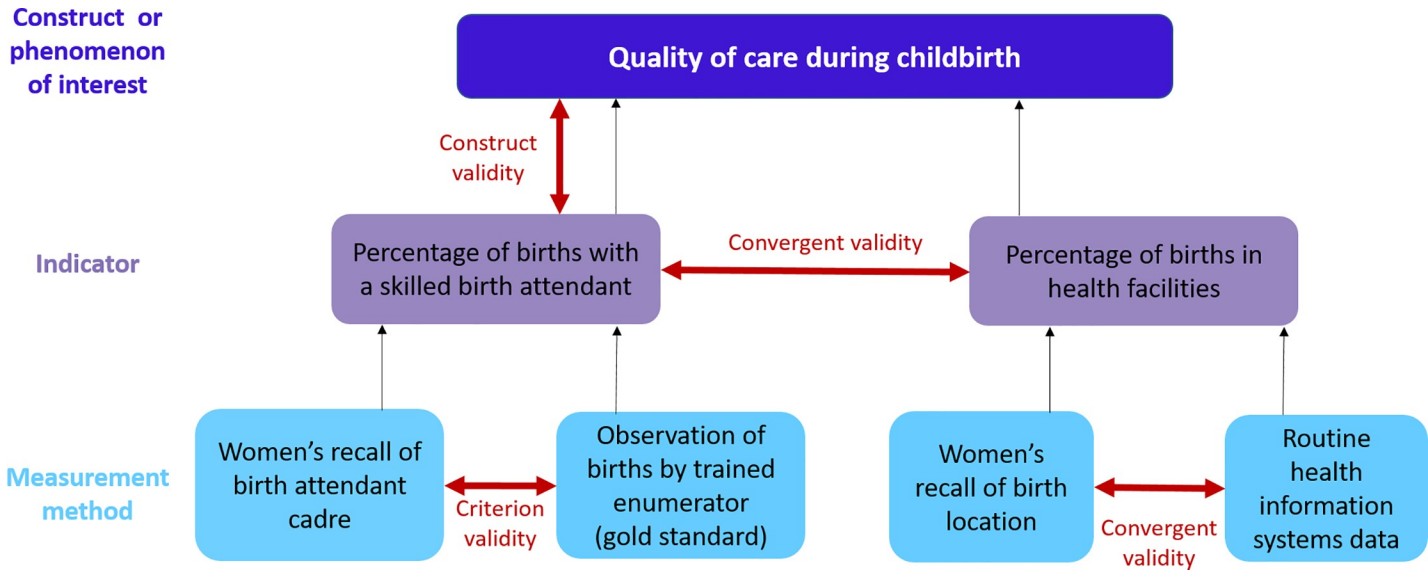

**Fig 3. Example of the three types of validity in maternal and newborn health indicators.**

c. Construct validity: An assessment of construct validity examines whether a given operalization (through indicator definition and its measurement) accurately reflects the phenomenon it is intended to measure. Construct validity is an umbrella term which subsumes all other types of validity, and therefore available assessments of criterion, convergent and other types of validity should be taken into consideration when evaluating the overall level of construct validity of an indicator.

## Criterion validity

Studies of maternal and newborn health indicators assessing criterion validity seek to understand the accuracy of a method of measurement compared to a "gold" or reference standard. Assessments of criterion validity measure the extent to which a current or proposed method of generating an estimate of an indicator accurately reflects an objective truth. Several key informants suggested that criterion validity, meaning the comparison of a measurement method to a gold standard, is perhaps the most commonly shared understanding of validity among the various stakeholders in the maternal and newborn health field. However, they also acknowledged that it captures the narrowest, most technical, aspect of indicator validity. Within maternal and newborn health, studies of criterion validity have predominantly assessed concurrent rather than predictive validity. The focus of criterion validity assessments has been largely on indicators of care coverage and content and to some extent on impact indicators. Examples of studies assessing criterion validity of maternal and newborn health indicators are shown in Table 1.

Many key informants noted that a substantial portion of recent work on assessing criterion validity has focused on indicators of care coverage and content captured in household surveys such as the Demographic Health Surveys (DHS) and Multiple Indicator Cluster Surveys (MICS). [52, 53] Munos and colleagues discuss many considerations and elements of diagnostic-style (criterion) validity studies related to assessing the validity of care coverage indicators based on data from population-level surveys. [42] A common approach to assessing validity of women's recall of specific events or care content is to compare women's recall (captured during an exit interview or sometime later during a home visit) against a "gold standard" based on

**Table 1.  Examples of studies assessing criterion validity.**

| Indicator domain | Examples of studies (reference standard and setting) |
|---|---|
| **Care coverage**<br><br>Coverage of maternal and newborn care interventions, skilled provider at birth, caesarean section rate | Women's self-report compared to:<br>• Observations during antenatal, intrapartum and postnatal care from<br>  ˚ Hospitals in Kenya [40, 43],<br>  ˚ Health facilities in Mozambique [44].<br>Women's self-report compared to:<br>• Medical records from medical booklets and HMIS: China through population-based survey [45]. |
| **Care content and quality**<br><br>Immediate initiation of breastfeeding, newborn resuscitation | Women's self-report compared to:<br>• Observations of care in hospitals in Tanzania, Bangladesh, Nepal [46]<br>• Medical records/facility registers in hospitals in Tanzania, Bangladesh, Nepal [46]. |
| **Impact**<br><br>Prevalence of maternal morbidities or obstetric complications (e.g. haemorrhage, pre-eclampsia/eclampsia, labour dystocia, prolapse), prevalence of severe maternal morbidity or near miss | Women's self-report compared to:<br>• Biomarkers and/or medical diagnoses as captured in clinical notes in samples drawn from:<br>  ˚ Indonesian hospitals [47],<br>  ˚ Ghana hospital [48],<br>  ˚ A tertiary hospital in Brazil [49],<br>  ˚ Maternity hospitals in Bolivia [50].<br>Women's self-report compared to:<br>• Clinical examination findings from a community survey in Egypt [51]. |

direct observations of care or, less commonly, care elements documented in a facility register or patient record. The most important quantitative metrics used by assessments of criterion validity are summarised in Table 2.

**Table 2.  Commonly used measures of criterion validity (adapted from [42, 54]).**

| Measure | Definition, calculation, meaning |
|---|---|
| **Individual-level validity** | |
| Sensitivity | The percentage of individuals with the outcome/characteristic of interest who were correctly classified as such. |
| Specificity | The percentage of individuals without the outcome/characteristic of interest correctly classified as such. |
| Percent agreement or Accuracy | The percentage of individuals who were correctly classified, i.e. for whom the outcome/characteristic of interest being measured is a match to the gold-standard comparison. |
| Positive predictive value | The probability that an individual who reported having an outcome/characteristic of interest truly had it. |
| Negative predictive value | The probability that an individual who did not report having an outcome/characteristic of interest truly did not have it. |
| Area under the receiver operating characteristic curve (AUC) | Plot of sensitivity versus 1-specificity. A value of 1 means a perfect match, 0.5 a random guess. For binary measures, this is the average of sensitivity and specificity. |
| Other, less commonly used, measures include likelihood ratio [49] and efficiency [48]. | |
| **Population-level validity** | |
| Inflation factor (IF) or Test to Actual Positive (TAP) ratio | Ratio of the population prevalence based on the measure being assessed in comparison with the true prevalence based on the gold standard. [55] This measure expresses the extent to which the true population prevalence of the indicator is under- or over-estimated, given the sensitivity and specificity of the measure under consideration and the true population prevalence. It is possible for an indicator to show low individual-level accuracy but good population-level accuracy. |

Some of the limitations of these predominantly facility-based criterion validity studies include limited generalisability, additional assumptions required to assess the extent of bias affecting population-level estimates, and issues with high coverage of routine care elements, which lead to sample sizes too small to calculate specificity. In addition, maternal and newborn health indicators based on population-level surveys have a two- to five-year recall period. Indicator validity is dependent on the ability of women to recall an event, which may be affected by length of time since the event. Only a few studies have assessed criterion validity based on length of the recall period since pregnancy and childbirth; many report substantial issues in the ability to ensure high follow-up rates and found some deterioration in the accuracy of women's recall as the length of recall period increases. [43, 44, 47, 49]

Despite the numerous metrics to statistically assess criterion validity, there is no consensus on what thresholds indicate acceptable or good indicator validity levels. Key informants agreed that there is no objective or recommended cut-off point for a "good" level of diagnostic validity that could single-handedly inform a recommendation to endorse the use of an indicator. Such endorsement would rely on crucial additional considerations, such as the intended use of the proposed indicator, quality of the data and its source(s), and quality of the gold standard used to assess validity. One key informant commented that "acceptable validity depends on how much imperfection you are willing to put up with and what purpose is the information for".

We present examples of pre-specified cut-offs provided by studies assessing validity of indicators based on women's recall (Table 3). It is important to note that most studies focus solely on assessing validity of indicator numerators. The validity of an indicator's denominator also has implications for the validity of the overall indicator, but has been less commonly evaluated. This is particularly important for indicators where the denominator is the population in need of an intervention. Decades of work to try to define the need for caesarean section as a denominator for a caesarean section rate indicator (including setting benchmark levels of caesarean section rates for all births irrespective of need used as a denominator) have led to the conclusion that the population of women in need of a caesarean section must be defined locally based on the epidemiological profile and context. [56, 57] Similarly, ongoing work to define appropriate denominators of newborns in need of targeted interventions such as resuscitation face a similar challenge since the population of newborns in need of resuscitation may vary based on different context and settings, e.g. be higher in referral compared to primary facilities. [58]

Key informants also highlighted the recent development and use of new indicators, such as those capturing maternal and newborn health financing, policies, and health system aspects. For health systems indicators, the validity of indicators capturing the existence of specific policies is sometimes referred to as "verification". Methods for such research might include a Ministry of Health representative reporting on policies, compared to the "gold standard" of policy existence as a ratified document, assessed through a document review. [59] Existence of a policy, however, does not guarantee its rollout or implementation, merely its existence.

## Convergent validity

The second common type of indicator validity assessment we identified in the literature compares estimates from various data sources or measurement approaches seeking to measure the same construct to understand the convergence between them (Fig 3). Studies assessing convergent validity, also referred to as "triangulation" by several key informants, aim to quantify the extent to which two or more estimates which should be related because they converge on the same theoretical construct, are in fact related. Assessments of convergent validity are commonly used in situations where a "gold standard" does not exist or is infeasible to estimate. A typical question asked in assessments of convergent validity is the extent to which a new/

**Table 3. Examples of pre-specified acceptable validity levels.**

| Source | Indicator types | Metric and Level |
|---|---|---|
| Ronsmans 1996 [54] | Maternal morbidity, obstetric complications | Fairly accurate if sensitivity and specificity >80%; high specificity is very important for rare outcomes to limit over-reporting of actual prevalence |
| Liu et al. 2013 [45] | Coverage and content of antenatal, delivery and postnatal care | Sensitivity/specificity:<br>• Low <0.33<br>• Moderate 0.33–0.66<br>• High >0.66<br>AUC–overall validity "high" if AUC>0.67 (otherwise moderate/low)<br>Population-level bias<br>• Small if 0.8<TAP ratio<1.2<br>• Moderate 0.5<TAP ratio<1.5<br>• Large <0.5 TAP ratio >1.5 |
| Stanton et al. 2013 [44] | Maternal and newborn health interventions during peripartum period in health facilities | Acceptable if<br>AUC>0.60 or IF 0.75–1.25<br>(Suggest indicators warranting incorporation into population-level surveys should meet both criteria- individual and population level validity) |
| Blanc et al. 2016 [40], McCarthy et al. 2016 [43] | Quality of maternal and newborn health care during childbirth | Individual-level accuracy measured by AUC<br>• High: AUC>0.7<br>• Moderate: 0.6<AUC<0.7<br>• Low: AUC<0.6<br>Degree of bias measured by IF<br>• Low 0.75<IF<1.25<br>• Moderate 0.5<IF<1.5<br>• Large IF<0.5 \| IF>1.5<br>High overall performance: high AUC + low IF<br>Reliability (decay in accuracy between baseline taken during exit interview and at 13–15 months post delivery)–Phi coefficient ($r_{phi}$)<br>• Poor if <0.4, Moderate between 0.4 and 0.6; High 0.6–0.8; > = 0.8 almost perfect agreement. |
| Blanc et al. 2016 [60] | Skilled birth attendant and key elements of maternal, intrapartum, newborn and immediate postnatal care among women with vaginal deliveries | Benchmarks of validity:<br>• AUC> = 0.6<br>• 0.75<IF<1.25<br>Overall acceptable indicator performance: both AUC and IF benchmarks met |
| Munos et al. [42], Chang et al. 2018 [61] | Coverage indicators | AUC≥0.70 |
| | | IF 0.75–1.25 |
| | | (study authors acknowledge these are arbitrary) |

AUC: Area under the receiver operating characteristic curve; TAP: test to actual positive ratio; IF: inflation factor.

different data source or estimation method compares to an established source or method. Studies also seek to understand the strengths and limitations, including financial feasibility, of the measurement approaches being compared. A wide range of methods has been used to examine the extent of agreement between distinct measurement methods and data sources used to calculate an indicator, including whether the data is on an individual, cluster (e.g. region, facility), or population level. Similarly to criterion validity, the cut-off point for an acceptable level of convergent validity is also subjective. Examples of studies assessing

**Table 4. Examples of approaches to assess convergent validity.**

| Indicator | Estimation method 1 | Estimation method 2 | Comparison |
|---|---|---|---|
| Stillbirth rate, neonatal mortality rate [62] | Full history of all live births and questions on pregnancies in the last five years resulting in non-live births | Full history of all pregnancies and their outcomes | Crude and adjusted risk ratios (determinants and clustering) |
| Maternal and perinatal mortality [63] | Enhanced community-level surveillance system | Routine data | Comparison of rate/ratios |
| Postnatal care coverage [64] | DHS questions | MICS questions | Descriptive comparison of proportions and timing |
| Caesarean section rate [65] | Population-based survey (DHS, MICS) | Health facility records | Linear regression coefficient; confidence interval overlap |
| Antenatal care coverage and content [66] | Individual-level data from clinical records weighted to population level | Aggregate routine health information systems reports | Simple comparisons of proportions and 95% confidence intervals |
| Estimates of value of aid for RMNCAH [67] | Comparison of estimates from four initiatives: Countdown to 2015, the Institute for Health Metrics and Evaluation, the Muskoka Initiative, and the Organisation for Economic Co-operation and Development (OECD) | | Simple differences |

DHS—Demographic Health Surveys, MICS—Multiple Indicator Cluster Surveys, RMNCAH–Reproductive, maternal, newborn, child and adolescent health.

convergent validity are shown in Table 4. We did not identify studies of discriminant validity (assessments of the extent to which an indicator is not associated with indicators or constructs it should not be associated with).

## Construct validity

One of the most important types of indicator validity highlighted in key informant interviews was construct validity. An indicator provides a simplified way of capturing a more complex phenomenon. Construct validity can be defined as the accuracy of the operationalisation of such phenomenon, and thus assesses the extent to which inferences can be made from the operationalization of an indicator to the theoretical construct which those operationalizations were intending to reflect. [68] In other words, the question is not how valid an indicator is, but how valid is this specific measurement of an indicator, in this place, at this time. In regard to indicator construct validity Arnold and Khan call this process of transforming concepts into indicators and further into survey questions the "validity of question". [69] The purpose of an indicator is central to assessing construct validity as well as other types of validity, [70] or, as noted by Etches and colleagues, "[a] concept-driven selection process should result in more methodologically sound indicators." [71]

The importance of clearly understanding and articulating an indicator's purpose was highlighted in a recent paper by Radovich and colleagues that examined the indicator capturing the percentage of births occurring with the assistance of a skilled birth attendant (SBA). [72] Several respondents emphasized that the process of assessing whether an indicator is "valid" should start with an understanding of not only the construct or phenomenon an indicator intends to measure, but also for whom and why. This includes a consideration of whether the underlying phenomenon itself is meaningful, that is, whether its purpose is important to maternal and newborn health and clearly understood by all stakeholders (S1 Table). A rigorous and complete assessment of construct validity must include both theoretical and empirical approaches, ideally involving the users of indicators for decision-making in the various global settings. [73] Yet, despite the importance attributed to construct validity by key informants, there was comparatively little published literature within maternal and newborn health focusing on this topic.

Some of the key indicators currently in use in the field of maternal and newborn health were developed, or are being used, as proxies for constructs that are considered important by

stakeholders but are not feasible or possible to measure simply or directly. This relates to, for example, maternal mortality (large sample sizes required means measurement is expensive) [74–76] and quality of care (a multi-dimensional construct requiring data on technical and clinical levels as well as patient's experience). [77] In particular, many key informants highlighted the importance of recent work around indicators of care content and quality, which concerned with the extent to which measurement methods can capture complex, multi-faceted constructs. Examples of this type of validity research, which also include considerations of face and content validity of measurement approaches (e.g. scales and questionnaires), include indicators of quality of care from a woman's perspective, [78] from a health facility perspective [79] indicators of complex care processes (e.g. case management of pre-eclampsia), indicators of autonomy and respectful care [80] and person-centered maternity care. [81] Additional challenges exist with measuring quality of newborn care, starting with the data source (newborns have limited communication and if the baby is taken out of the mother's sight, she cannot report accurately). [82] While this work forms a large part of the current validation research of maternal and newborn health indicators, it is not yet fully formed.

## Discussion

Using mixed methods, we identified three common types of indicator validity used in the field of maternal and newborn health, all of which have a role in evaluating the performance of indicators. Key informant interviews revealed that a variety of definitions and interpretations of indicator validity exist, highlighting the need to establish a common language and understanding of indicator validity among global and local maternal and newborn health stakeholders. We have attempted to synthesize key concepts and to present a typology of indicator validity that characterizes the varied ways in which the concept of validity is understood and assessed in the literature, indicator guidance documents and by a sample of maternal and newborn health stakeholders. We suggest that those who develop, assess or recommend maternal and newborn health indicators clarify their understanding of the various types of validity of studied or recommended indicators.

Despite the importance of construct validity highlighted in key informants' responses, we identified a gap in the literature and indicator guidance documents in explicitly describing and evaluating the underlying phenomena which various maternal and newborn health indicators seek to measure, and an absence of studies of construct validity in general. For example, is the SBA indicator intended to measure an enabling childbirth health care environment, coverage of good quality childbirth care, minimum safety levels during childbirth, to be a proxy for maternal mortality, or relate to multiple constructs? Conceptual understanding of the underlying phenomena that specific indicators are intended to measure may vary across stakeholders using the indicators and may change over time, but are rarely made explicit.

There is a predominance of validation studies on the narrowest conceptualisation of validity–criterion validity–but the larger issue of the construct and its meaning for progress in maternal and newborn health is rarely addressed. Once developed, used and measured with a high uptake for many years, maternal and newborn health indicators tend to remain in use for decades. However, the constructs being measured by such indicators are often unclear or may evolve in importance over time. We also highlight a view shared by many key informants that an indicator's performance on assessment of criterion validity should not be the sole determinant of its use for monitoring and decision-making; its measurement parameters need to be "good enough" for the purpose at a given time and place. [25] One such aim could include generating aspirational indicator estimates for the purpose of improving quality of data or measurement methods for the future. [8]

There is a growing concern with the large number of maternal and newborn health indicators used across several initiatives, including the variation in indicator definitions and the resources required to produce such indicators. [6] A more consistent understanding of indicator validity could help guide the prioritization, development and testing of more robust maternal and newborn health indicators. [14] Improved global coordination among stakeholders conducting or supporting validation studies is needed to avoid duplication of efforts. Further, it is crucial to consider the perspectives of country-level stakeholders in prioritising which types of validation matter most for which indicators and which types of indicators should be validated first and where. The development of guidance and criteria for assessing common types of indicator validity, linked to an action plan to prioritize indicators for validation, could help improve such coordination. Coordinated research to assess validity of a smaller number of locally relevant core indicators that seek to measure important constructs could help accelerate action to improve maternal and newborn health. In parallel, it is also vital to coordinate assessment of indicator validity with assessment of other important attributes of indicators, including feasibility and reliability. [83] Studies which describe elements of clarity, feasibility and acceptability of data collection tools, [84–87] such as those employed in qualitative studies and cognitive interviewing, [84] are complementary to other assessments of validity.

## Limitations

We used a literature review and key informant interviews to explore the field of indicator validation research in maternal and newborn health indicators. We conducted a comprehensive review of the literature published in English since 1990 to identify key themes and provide examples and acknowledge that our review may have missed relevant publications in languages other than English. We also acknowledge that while the key informants included measurement experts and authors of many of the recently conducted validation studies on maternal and newborn health indicators within the maternal and newborn health field, our sample of key informants included only English-speaking respondents working predominantly at the global level and did not include many country-level experts and stakeholders. We did not aim to summarize the findings of all validation studies for individual indicators; however, such systematic reviews and meta-analyses could be a useful next step for summarising the available evidence.

While we were informed by "multidisciplinary bodies of knowledge" which are needed for high quality conceptual frameworks, it is important to recognise that the issues surrounding validity of population health indicators are somewhat different from those of tools or questionnaires as elaborated in other disciplines, particularly psychology. [88] Some distinct types of validity used in these fields are not relevant to our topic and the definitions of validity we propose in this framework do not completely overlap with definitions used in other disciplines.

## Conclusion

Indicator validation is a part of a continuous process of building and synthesising evidence on indicator performance. We found that in the maternal and newborn health literature and among measurement experts, the term validity is used broadly to capture a variety of indicator performance assessments. Some of the current challenges related to harmonization and coordination of maternal and newborn health indicators stem from a heterogeneity of definitions of indicator validity, often by stakeholders from various disciplinary backgrounds. We recommend that the language used to describe validation research should be more precise as to the specific type(s) of validation assessed and the related findings (e.g. an indicator described as "valid" or "validated" should be nuanced and time- and context-specific).

In addition to the three most common types of maternal and newborn health indicator validity identified, we highlight the fact that any appraisal of an indicator's validity requires clarity about the construct that the indicator is intending to measure. We therefore recommend that future initiatives to coordinate indicator validity research focus on important underlying constructs rather than individual indicators (which represent the operationalization of constructs). This approach can help align stakeholders to develop a clear understanding of how best to measure important constructs, including agreement on "how not to measure" a construct for which "valid" indicators may not yet have been developed and tested.

## Supporting information

**S1 Table. Overview of key response themes from interviews with key informants.** (DOCX)

## Acknowledgments

The authors would like to acknowledge the key respondents' participation in interviews and discussions with members of the MoNITOR technical advisory group.

## Disclaimer

This report contains the collective views of the authors and does not necessarily represent the decisions or the stated policy of the World Health Organization.

## Author Contributions

**Conceptualization:** Lenka Benova, Ann-Beth Moller, Allisyn C. Moran.

**Data curation:** Lenka Benova, Ann-Beth Moller, Kathleen Hill, Lara M. E. Vaz, Alison Morgan, Claudia Hanson, Katherine Semrau, Allisyn C. Moran.

**Formal analysis:** Lenka Benova, Ann-Beth Moller, Kathleen Hill, Lara M. E. Vaz, Alison Morgan, Claudia Hanson, Katherine Semrau, Shams Al Arifeen, Allisyn C. Moran.

**Funding acquisition:** Ann-Beth Moller, Allisyn C. Moran.

**Methodology:** Lenka Benova, Ann-Beth Moller, Kathleen Hill, Lara M. E. Vaz, Alison Morgan, Claudia Hanson, Katherine Semrau, Shams Al Arifeen, Allisyn C. Moran.

**Supervision:** Ann-Beth Moller, Allisyn C. Moran.

**Visualization:** Lenka Benova, Ann-Beth Moller, Kathleen Hill, Lara M. E. Vaz, Alison Morgan, Claudia Hanson, Katherine Semrau, Shams Al Arifeen, Allisyn C. Moran.

**Writing – original draft:** Lenka Benova, Ann-Beth Moller, Kathleen Hill, Lara M. E. Vaz, Alison Morgan, Claudia Hanson, Katherine Semrau, Shams Al Arifeen, Allisyn C. Moran.

**Writing – review & editing:** Lenka Benova, Ann-Beth Moller, Kathleen Hill, Lara M. E. Vaz, Alison Morgan, Claudia Hanson, Katherine Semrau, Shams Al Arifeen, Allisyn C. Moran.

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
