## [Decision Letter · Decision Letter 0]

10 Dec 2019

PONE-D-19-29431

What is meant by validity in maternal and newborn health measurement? A conceptual framework for understanding indicator validation

PLOS ONE

Dear Dr Benova,

Thank you for submitting your manuscript to PLOS ONE. After careful consideration, we feel that it has merit but does not fully meet PLOS ONE’s publication criteria as it currently stands. Therefore, we invite you to submit a revised version of the manuscript that addresses the points raised during the review process.

ACADEMIC EDITOR: The paper is well-written and has adequate scientific value. However, there was a general agreement among reviewers about the need to improve the explanations of the different types of validity discussed. 

We would appreciate receiving your revised manuscript by Jan 24 2020 11:59PM. To enhance the reproducibility of your results, we recommend that if applicable you deposit your laboratory protocols in protocols.io, where a protocol can be assigned its own identifier (DOI) such that it can be cited independently in the future. For instructions see: http://journals.plos.org/plosone/s/submission-guidelines#loc-laboratory-protocols

We look forward to receiving your revised manuscript.

Kind regards,

Abraham Salinas-Miranda

Academic Editor

PLOS ONE

Journal Requirements:

1.

2. We note that you have stated that your data are currently in the process of being deposited . Should your manuscript be accepted for publication, we will hold it until you provide the relevant accession numbers or DOIs necessary to access your data. If you wish to make changes to your Data Availability statement, please describe these changes in your cover letter and we will update your Data Availability statement to reflect the information you provide.

Additional Editor Comments (if provided):

The paper has been reviewed by independent reviewers who agree with the value of the manuscript for the readership of PLOS ONE and the important contribution to the literature. Hence, the recommendation is to accept upon minor revisions. Given the comprehensive nature of the manuscript that required diverse expertise, the Academic Editor requested more than the minumum number of reviewers and invited reviewers with different expertise (in maternal and child health, measurement, and epidemiology). The reviewers generally agreed with the need to make some changes in the manuscript, particularly related to the definition of validity and measurement validity. Please make the indicated corrections and resend a revised version.

Reviewers' comments:

Reviewer's Responses to Questions

**Comments to the Author**

1. Is the manuscript technically sound, and do the data support the conclusions?

Reviewer #1: Partly

Reviewer #2: Yes

Reviewer #3: Partly

Reviewer #4: Yes

Reviewer #5: Yes

Reviewer #6: Yes

2. Has the statistical analysis been performed appropriately and rigorously? 

Reviewer #1: N/A

Reviewer #2: No

Reviewer #3: N/A

Reviewer #4: N/A

Reviewer #5: N/A

Reviewer #6: N/A

3. Have the authors made all data underlying the findings in their manuscript fully available?

Reviewer #1: Yes

Reviewer #2: Yes

Reviewer #3: No

Reviewer #5: Yes

Reviewer #6: Yes

4. Is the manuscript presented in an intelligible fashion and written in standard English?

Reviewer #1: Yes

Reviewer #2: Yes

Reviewer #3: Yes

Reviewer #4: Yes

Reviewer #5: Yes

Reviewer #6: Yes

5. Review Comments to the Author

Reviewer #1: The topic covered by this paper is well timed, and well needed. How validity is defined, and ultimately what should be considered a valid maternal and newborn health indicator is important. As such, the paper has potential to provide much needed guidance in the field, and help to focus the 140+ MNCH indicators currently in play.

However, the authors would benefit from setting the stage early of what is meant by validity, and if they intend to also talk about reliability. This would mean a thorough (though concise) review of the definitions of validity, anchored with multiple strong referrences published in peer reviewed journals or text books. This will help readers also know what to look for when reading the author's results, and may help the authors taylor the way they present their results, key points they highlight, and what they ultimately discuss. There are a couple of types of validity - namely face validty, and content validity, both important, that are missing. 

The definitions of criterion, construct, and convergence validity are also wrong or incomplete. Here are a couple of sources quickly found online:

https://socialresearchmethods.net/kb/convdisc.php - helpful for all validity discussions re def)

https://www.scribbr.com/methodology/types-of-validity/

In addition, the authors could explicitly state whether or not some indicators can be judged as "more valid" than others if they meet more than one validity criteria. Ideally, a few examples could be provided of indicators and which criteria they meet. 

Until the definitons of validity are placed well, the authors will find it difficult to concretely discuss the range of how validity is defined within MNCH, and (hopeffully) highlight key gaps if they exist.

The authors also need to clarify the relationship between reliability and validity. While it is true that an indicator can be reliable wtihout being valid, this is not a reciprical relationship: an indicator cannot be valid wtihout being reliable. Please see here: https://wonderlic.com/validity-and-reliability/

The authors did a great job reviewing teh MNCH literature, but they are also encouraged to review the validity literature, align their definitions of the various types of validity with those commonly understood within the literature, and then review their data and update their analysis in light of these revised definitions of validity.

Overall on the analysis, since all data was qualitative, it would be useful to highlight key points with an occasional quote, if possible, or demonstrate a sense of whether or not certain opinions are those of only one person or multiple people. Key informants do not seem to contribute much to the discussion of criterion or convergent validity, though it would strenghten the study to integrate a bit more from Key informants on these topics.

This paper has immense potential. I hope these observations are useful.

Reviewer #2: The paper presents an important perspective on how validity of maternal and newborn indicators can be defined and understood. This is important especially considering that the definitions and data sources used to produce indicator estimates vary and challenges exist with completeness, accuracy, transparency, and timeliness of data in many LMICs. Generally, the paper is well written and a few points to consider are as below:-

Introduction:

- Well summarized including the global and regional burden of the problem highlighting LMICs.

- References used are up-to-date

Materials and Methods:

- Overall good structure followed

- Interviews were conducted by one interviewer between December 2017 and November 2018. Were the interviews conducted face-to-face, by telephone or skype?

- One interviewer conducted the data collection from the 32 participants over a period of time. Can the authors indicate why recording the interviews was not opted for, yet it is an easier and faster way of collecting qualitative interviews and minimizes recall bias from the interviewer perspective? In addition, it would be nice for the authors to summarize the backgrounds of the participants interviewed.

- Only one interviewer did the conducted and performed the shorthand note taking between 45 – 90 minutes. How was this structured?

- How was the data from the 32 interviewees analysed? There is no analysis plan in the manuscript.

- Ethics – data collected involved interviewing human subjects on the subject matter. Why was ethical approval not sought by the authors?

Results and Discussion

- Authors have reported the results in a good style with sub-headings separating the various aspects.

- It is important that the authors highlighted the issue of subjectivity in the acceptable levels of validity in the criterion and convergent validity.

- Main points have been well summarized in the discussion section

References

- Robust review of literature done

- Authors to review references 1,2,4,17,19 and 24 to include the URLs/web links and access dates as these are online/electronic sources as per the recommended citation style.

Reviewer #3: The topic addressed is an important area in which some confusion is common, in general, ie not limited to maternal health. So, this manuscript has potential to make a useful contribution to help clarify issues within this field. However, there are two aspects which I think need improvement to strengthen this work:

i) the three types of validity discussed belong within an established theoretical framework. I recommend that this be acknowledged more explicitly, by providing the definitions of these in the introduction rather than in the results, perhaps with a little more reference to the diversity of types of validity considered in literature.

ii) The methods indicate that a literature review was conducted (line 117). Although the methods for identification of literature are stated (lines 131-8) the results provide very limited information regarding the literature identified, not even the number of manuscripts identified is mentioned. It is for this reason that I have indicated that all data underlying finding are not made fully available. More details on the findings should be provided.

Other minor matters for attention:

Line 169 - refers to S2 Table, but I could not locate it

Line 186 – surely multiple measurements need to be compared with the relevant ‘gold standard’ measurements to examine criterion validity.

Lines 192-3 - please clarify how the statement is related to the concept of convergent validity.

Lines 247-50 - examples are useful but please provide a methodological reference for lines 247-8.

Lines 223-5 - this statement does not appear to be supported within the examples provided in Table 1.

Line 303 - more detail of the indicator needs to be provided: birth attended by ? Are lines 303-4 referring to source 58 or to the interviewees of the reported study?

Lines 341-3 - several potential uses of the indicator are mentioned. It is however unclear why the discussion seeks to align the indicator with only one use, when in fact it may fulfil each of the stated uses in differing contexts.

Lines 353-5 - the intended meaning is unclear.

Line 479 - journal details for ref [26] need to be corrected.

There are a number of occurrences of the term ‘this paper’, some of which could be dropped.

Tables and Figures

I found it confusing that text refers to Tables and Figures, but they are entitled Tabs and Figs

It is unclear what is plotted in each sub-plot in Figure 2 – what do the two axes represent? In line 206 reference to hitting the bullseye is confusing since the plot for high validity and low reliability has points scattered around the bullseye but spread across the entire circular region.

In table 1 the headings could be clearer, eg in the second column the content is: “Indicator: gold standard / setting (study/ies”); since impact indicators are reported to be less common it is surprising that they appear first in the table.

In Table 2 the definitions of predictive values are too narrow: these terms are used within diagnostic assessments and thus revolve around a test providing a result rather than an individual reporting something. Additionally the definition for AUC does not actually state how AUC is derived, it focuses more on the definition of the ROC (receiver operating characteristic) curve.

Table 3 – the text indicates that it is about indicators in population-based surveys. This should be reflected in the title of the Table.

Table 4 provides limited details about the data used and the methods of comparison. For example for Caesarean section the source reference uses both coefficients obtained using linear regression, for multiple studies, and also provides a scatterplot which displays the data. By contrast for RMNCAH is for just one study and each of the four methods of estimation for a number of indicators are reported, which enables simple differences to be derived, though they are not actually derived.

Reviewer #5: Thank you for the opportunity to review the manuscript.

The authors should be congratulated on a well-written and timely paper presenting a range of perspectives on how validity of maternal and newborn indicators is defined and understood by those who develop and use these indicators. The study has important implications for ongoing global efforts to track maternal and newborn health progress. Please see below for specific comments on the manuscript:

Materials and methods

• Line 117: Please reference the key indicator guidance documents used in the review.

• Line 22: Can the authors please provide further details about the key informants’ backgrounds/expertise on this topic or in MNH? The authors provide some background information in the limitations section of the discussion however it would be useful to have this outlined earlier. Were all key informants’ part of the MoNITOR group?

• Was there any pre-testing of the interview guide prior to the 32 key informant interviews?

• What was the consent process?

• Reference 23: Is this paper accepted for publication or under review? Please update reference.

• Line 127: What written materials and relevant publications were received from the respondents after the interview? How were these relevant to the interviews?

• Can the authors please clarify the process for the qualitative data analysis? Apologies if I have overlooked this. Was the analysis undertaken by the same person who undertook the interviews?

Results

• Line 223: The authors state that key informants noted that a substantial portion of recent work on assessing criterion validity has focused on indicators of care coverage. It would be useful to know what proportion of the key informants identified this to be the case.

The discussion is well-written, and the conclusion is concise and logical. This is an interesting and useful paper that should be considered for publication following revision of comments.

Reviewer #5: Review Comments

This is a very interesting article; however there are some aspects that need to be incorporated in some sections in order to improve the manuscript and comprehension, especially regarding methods. A more detailed descriptions in some sections are required.

Reviewer #6: Review Comments

In general terms, the present study aims at gathering concepts and themes essential when analysing indicator validity in the context of maternal and newborn health indicators with the goal of reaching high health standards of these two population groups. Personally, I believe the main subject of this original body of work made a substantial contribution in presenting the existing knowledge about indicator validity through a qualitative investigation based on interviews to key informants, literature revision and which had the participation of the MoNITOR group. The results of the investigation are a tool that will allow all stakeholders in maternal and newborn health to reconsider the ongoing work about indicator validity. This work will improve long- term health of pregnant women and their newborns who are a vulnerable group through their life course.

Introduction:

In this section, reference is made to the background and settings in which this work is developed, objectively contributing to filling the information gap existing when discussing indicator validity in maternal and newborn health. Consequently, the objective of this study was clearly stated in this section.

I suggest reviewing the content in parentheses in line sixty-eight.

Materials and methods:

This section includes an organized structure in which the study design, sample and type of sample for the selection of key informants are stated. It also explains, in detail, the terms used to conduct the review and it sets the limits of maternal and newborn health indicators. Moreover, it incorporates a definition of concepts.

Since this work was developed in four stages, I suggest mentioning the four stages in the first paragraph of the section. I also suggest, moving the interview to key informants to the beginning adding that the interview is part of a qualitative study which main result and methodology were already published. Then, I would list the review and MoNITOR group discussion. I give this advice considering that is the way the rest of the section is organised.

The second paragraph begins with the type of sample used to select key informants and, even though it is indicated that the first results of these interviews were previously published in another study, I think it is necessary to add the main details of the methods of such work in this section of this study.

Ethics Approval:

I suggest adding, at the end of Materials and Methods, ethical aspects of the study, such as whether the researchers had informed consent of key informants and why they did not present the protocol before the ethics committee.

Results:

The results are in accordance with the objects of the study, but in the case of the review the results are not complete because they do not inform the number of studies obtained through the research nor the studies selected. I suggest completing with these pieces of information.

The first paragraph of this section mentions Table S2 (line 168), but in the annexed material there is only a Table S1. Line 297 mentions S1, I suggest checking if this Table is indeed Table 1.

Discussion:

This section discusses the study’s main results. Some aspects are discussed in the section Results, for instance, when limitations of the use of the denominator in indicator validity are mentioned.

Conclusion:

It focusses in the objectives of the study, highlighting the main recommendations of the study.

Bibliographic references: 

• I suggest revising the references and writing them according to Vancouver style.

• I suggest to add the “doi” to all the citations.

• Review the reference number 20, seems to be have two references together.

• Standardize the name of the journals. Some journal´s names are written completely and others, abbreviated.

• Please, be sure you have the publication tittle name of all your citations, not only the link, and the date of access.

6. PLOS authors have the option to publish the peer review history of their article (what does this mean?). If published, this will include your full peer review and any attached files.

Reviewer #1: Yes: Rachel Jean-Baptiste, MPH, PHD

Reviewer #2: Yes: Duncan N. Shikuku

Reviewer #3: No

Reviewer #4: Yes: Lorena Binfa

Reviewer #5: No

Reviewer #6: No

---

## [Author Response · Author response to Decision Letter 0]

24 Jan 2020

We have provided responses as an attached document.

---

## [Decision Letter · Decision Letter 1]

4 May 2020

PONE-D-19-29431R1

What is meant by validity in maternal and newborn health measurement? A conceptual framework for understanding indicator validation

PLOS ONE

Dear Dr Benova,

Thank you for submitting your manuscript to PLOS ONE. After careful consideration, we feel that it has merit but does not fully meet PLOS ONE’s publication criteria as it currently stands. Therefore, we invite you to submit a revised version of the manuscript that addresses the points raised during the review process.

We would appreciate receiving your revised manuscript by Jun 18 2020 11:59PM. To enhance the reproducibility of your results, we recommend that if applicable you deposit your laboratory protocols in protocols.io, where a protocol can be assigned its own identifier (DOI) such that it can be cited independently in the future. For instructions see: http://journals.plos.org/plosone/s/submission-guidelines#loc-laboratory-protocols

We look forward to receiving your revised manuscript.

Kind regards,

Emma Sacks

Academic Editor

PLOS ONE

Editor Comments:

Hi Lenka,

Apologies for the delay (it looks like the previous handling editor became unavailable and your manuscript was without an editor for some time before it was assigned to me, but I've tried to accelerate this round of review and believe the last revisions should be relatively quick!).

This paper is very comprehensive, and an important analysis of the types of indicators available.

There are only some additional minor comments from two reviewers to address. In addition, if you could address the following, that would be helpful.

-Can you specify who the key informants were. I know there is reference to another paper will full methods, but it would be good to summarize in a sentence or so here.

-Can you clarify if no ethical approval was sought, or if it was exempted because considered to be non human subjects research? The paper includes a quote, so even though it's not sensitive personal information, many ethical review boards would want to review a protocol that includes interviews.

This is completely discretionary if you want to address in the discussion at all since you want to include both maternal and newborn health measurements, but because of my work on trying to measure experience of care for newborns (https://www.ncbi.nlm.nih.gov/pmc/articles/PMC5445465), I'm always aware of the additional challenge with measuring newborn quality of care because of who is reporting (if the baby is taken out of the room, the mother can't report and the health worker may not record etc etc.) since newborns have limited verbal communication as opposed to measures for women who can self-report.

Thanks so much.

Reviewers' comments:

Reviewer's Responses to Questions

**Comments to the Author**

1. If the authors have adequately addressed your comments raised in a previous round of review and you feel that this manuscript is now acceptable for publication, you may indicate that here to bypass the “Comments to the Author” section, enter your conflict of interest statement in the “Confidential to Editor” section, and submit your "Accept" recommendation.

Reviewer #1: All comments have been addressed

Reviewer #2: All comments have been addressed

Reviewer #3: (No Response)

2. Is the manuscript technically sound, and do the data support the conclusions?

Reviewer #1: Yes

Reviewer #2: Yes

Reviewer #3: Partly

3. Has the statistical analysis been performed appropriately and rigorously? 

Reviewer #1: Yes

Reviewer #2: Yes

Reviewer #3: N/A

4. Have the authors made all data underlying the findings in their manuscript fully available?

Reviewer #1: Yes

Reviewer #2: Yes

Reviewer #3: Yes

5. Is the manuscript presented in an intelligible fashion and written in standard English?

Reviewer #1: Yes

Reviewer #2: Yes

Reviewer #3: Yes

6. Review Comments to the Author

Reviewer #1: I reviewed comments I made on the earlier draft of this paper. Compared to that early draft, this paper is much, much better. I appreciate the authors taking into consideration the advise shared and incorporating.

The paper is a much better ps thaper, and I don't have any additional comments. I like their use of examples of indicators within each category of validity. It would have been useful to then show a table with some indicators that meet all validity criteria, as this would help the reader think about how to evaluate an indicator they choose to use with the aim of getting the most valid indicators possible (gold standard). It would be a useful contribution if the authors conclude with an example of a few indicators that are like 'gold standard' MCH indicators because they meet various validity criteria and all researchers are encouraged to include these in their studies. This would be similar to how demographics indicators always include age and gender.

Other than that, great work!

Reviewer #2: Overall, the authors appear to have satisfactorily addressed the comments and the manuscript is clearer and much improved. No further revisions are suggested at this time.

Reviewer #3: The relationship between validity and reliability raised by reviewer 1 is not adequately addressed. The authors assert that ‘theoretically an indicator can be valid but unreliable despite this being counterintuitive and uncommon’. No basis for this assertion is given. Streiner et al (2015) in their monograph “Health measurement scales: a practical guide to their development and use” (5th edition) explain that reliability places an upper limit on validity.

A previous article by Streiner and Norman is cited (39) in support of Figure 2. I do not have access to this article. However Streiner et al (p164) say “We have argued (Streiner and Norman, 2006) that the other terms ignore the equally important part of the denominator, “Subject variability”. . . .if two raters place all of their students in the ‘above average’ category, and do so again when asked to re-evaluate the students one week later, their repeatability, reproducibility, consistency and agreement will all be perfect, but the reliability will be zero, because there is no true difference among those being rated.” This example illustrates why the assertion is not well-founded.

Thank you for moving the definition of validation. However it is at the end of the materials and methods section, after numerous uses of the term. An earlier indication of the definition would be preferable.

In response to my comment on Table 3 it was indicated that “we opted to revise the text describing this Table”. However the only revision I see is deletion of reference to population-based surveys!

7. PLOS authors have the option to publish the peer review history of their article (what does this mean?). If published, this will include your full peer review and any attached files.

Reviewer #1: Yes: Rachel Jean-Baptiste Salomonsen, MPH, PhD

Reviewer #2: Yes: DUNCAN SHIKUKU

Reviewer #3: Yes: Sarah Ann White

---

## [Author Response · Author response to Decision Letter 1]

15 May 2020

We have attached the responses as a separate document.

---

## [Editor Report · Decision Letter 2]

18 May 2020

What is meant by validity in maternal and newborn health measurement? A conceptual framework for understanding indicator validation

PONE-D-19-29431R2

Dear Dr. Benova,

We are pleased to inform you that your manuscript has been judged scientifically suitable for publication and will be formally accepted for publication once it complies with all outstanding technical requirements.

With kind regards,

Emma Sacks

Academic Editor

PLOS ONE

Additional Editor Comments (optional):

Thanks for your thoughtful revisions, and important contribution to the literature.
---

## [Editor Report · Acceptance letter]

21 May 2020

PONE-D-19-29431R2 

What is meant by validity in maternal and newborn health measurement? A conceptual framework for understanding indicator validation 

Dear Dr. Benova:

I am pleased to inform you that your manuscript has been deemed suitable for publication in PLOS ONE. Congratulations! Your manuscript is now with our production department. 

With kind regards,

on behalf of

Dr. Emma Sacks 

Academic Editor

PLOS ONE